# Hepatic ROS Mediated Macrophage Activation Is Responsible for Irinotecan Induced Liver Injury

**DOI:** 10.3390/cells11233791

**Published:** 2022-11-26

**Authors:** Bohao Liu, Cong Ding, Wenbin Tang, Chen Zhang, Yiying Gu, Zhiqiang Wang, Tingzi Yu, Zhuan Li

**Affiliations:** 1The Key Laboratory of Model Animals and Stem Cell Biology in Hunan Province, Hunan Normal University School of Medicine, Changsha 410013, China; 2The Key Laboratory of Study and Discovery of Small Targeted Molecules of Hunan Province, Hunan Normal University School of Medicine, Changsha 410013, China; 3Department of Pharmacy, Hunan Normal University School of Medicine, 371 Tongzipo Rd, Changsha 410013, China

**Keywords:** irinotecan, liver injury, chemotherapy, macrophage activation, ROS

## Abstract

Irinotecan is the first line chemotherapy drug used for treatment of metastatic colorectal cancer worldwide. There is increasing evidence suggesting that liver damage, including steatosis and steatohepatitis, can be caused during the treatment involving irinotecan. However, molecular mechanisms by which irinotecan-induced liver injury remain elusive. In this study, we found that irinotecan treatment caused significant elevation of ALT, inflammation, and fat accumulation in the liver, which are associated with hepatic macrophage activation. Depletion of macrophages by clodronate liposome improved irinotecan induced liver injury and inflammatory response in mice. In vitro data indicated that irinotecan induced intracellular ROS production in primary hepatocyte and upregulating of toll-like receptor (TLRs) family expression in macrophages. Supernatant from irinotecan treated hepatocyte triggered macrophage activation and upregulation of TLRs in macrophage, and N-acetylcysteine (NAC) abolished these effects. By using co-culture system, we further revealed that irinotecan activated macrophage induced impairment of lipid metabolism and promoted apoptosis in hepatocyte and NAC prevented macrophage-induced cell death and partially revered impaired lipid metabolism in hepatocytes. By using the irinotecan liver injury model, we demonstrated that combining NAC with irinotecan prevented irinotecan-induced macrophage activation, TLR upregulation, liver injury, and partially prevented the accumulation of triglycerides in liver. Our results thus indicated that macrophages play a critical role in irinotecan-induced liver injury, and targeting ROS provides new options for development of hepatoprotective drugs in clinical practice.

## 1. Introduction

The topoisomerase I inhibitor irinotecan (IRT), a semisynthetic derivative of calprotectin, is a chemotherapeutic agent widely used in the treatment of metastatic colorectal cancer [1,2]. By inhibiting the DNA topoisomerase I complex and causing DNA double-strand breaks, irinotecan induced cytotoxicity to exert its antitumor effects. In the presence of carboxylesterase 2 (hCE2) [3], irinotecan is metabolized in the blood and liver to the active metabolite SN-38, which is then inactivated in the liver by conversion of uridine diphosphate-glucuronosyltransferase 1A1 (UGT1A1) to SN-38G (β-glucosidic acid conjugate) [4]. Irinotecan can nonspecifically damage rapidly proliferating cells, and common adverse effects include neutropenia and delayed diarrhea [5]. Mounting evidence suggests that irinotecan-involved chemotherapy causes hepatotoxicity, including steatosis and steatohepatitis that can lead to life-threatening situation, as well as discontinuation of chemotherapy in patients. Vauthey et al. [6] found that irinotecan is associated with steatohepatitis by analyzing 406 patients with colorectal cancer liver metastasis who underwent hepatic resection. Another study indicated that irinotecan-treated CRCLM patients have higher incidence of steatohepatitis [7,8,9]. Furthermore, irinotecan induces steatohepatitis and increases the risk of fibrosis, cirrhosis, and liver failure [10,11,12] and is gaining clinical attention. However, the specific molecular mechanism of irinotecan-induced liver injury remains elusive.

Steatohepatitis is a common form of liver injury in which fat aggregation (steatosis), hepatocyte damage, and death and inflammatory cell aggregation are observed histologically [13,14]. The pathogenesis of drug-induced steatohepatitis usually involves excessive hepatic fat deposition, mitochondrial dysfunction, and increased formation of reactive oxygen species (ROS) [15]. In the case of irinotecan-induced steatohepatitis, it is often accompanied by neutrophil infiltration, increased levels of ROS, and induced expression of tumor necrosis factor-α (TNF-α), IL-1α, IL-1β, and IL-6 [16]. Mahli et al. [17] reported that irinotecan induces activation of extracellular signal-regulated kinase (ERK)and impairs autophagic flux, which leads to cellular lipid accumulation by alkalinizing lysosomal pH in hepatocytes. In irinotecan-induced nonalcoholic steatohepatitis (NASH) mouse models, high expression of inducible nitric oxide synthase (iNOS) in mouse liver tissue is also observed [18]. Those data suggested that irinotecan-induced liver injury involves an impairment of hepatocyte function and inflammation, but the specific molecular mechanism underlying irinotecan-induced steatohepatitis remains unclear.

Macrophages are highly heterogenous and possess plasticity, which able them to develop distinct adaptation to slight alteration of tissue environments, including nutrients, metabolites, and oxygen, which results in significant diversity of macrophages even within the same tissue [19]. During the early period of inflammation, macrophages transit to pro-inflammatory (M1) types through classical activation pathways to defend intracellular bacteria or viruses. While, in late inflammation, macrophages polarize to alternative (M2) type through the alternative activation pathway to help tissue healing and to tolerate self-antigens [20,21]. Macrophages play important roles in maintaining the stability of the internal environment by removing pathogenic factors and cancerous cells [22]. ROS and pro-inflammatory cytokines, including IL-1β, IL-6 and TNF-α, are secreted, which contribute to pro-apoptotic activity of classically activated macrophages [23]. In fatty liver, the largest proportion of TNF-α is produced by macrophages, and depletion of macrophages results in significant reduction in TNF-α levels [24]. The inhibition of macrophage activation has a significant effect on the treatment of inflammation-related diseases [25,26,27]. Liu J et al. [28] found that stauntoside B, a C21 steroidal glycoside isolated from the Chinese medicine Baiqian, inhibits macrophage activation and acts as a potent NF-κB inhibitor, showing therapeutic effects in the treatment of inflammatory diseases. In addition, Peng et al. [29] found that a highly selective catalytic p300/CBP inhibitor is able to attenuate acute liver injury by modulating macrophage polarization and inhibiting inflammatory cytokines. However, the phenotype and function of macrophages in irinotecan-induced liver injury are not fully understood.

In this study, we investigated the role of macrophages in irinotecan-induced liver injury both in vitro and in vivo. Our data indicated that irinotecan caused liver injury by stimulating the activation of macrophages, and the clearance of macrophages improved the irinotecan-induced inflammatory response and liver injury. We further revealed irinotecan-induced reactive oxygen specie (ROS) accumulation in hepatocyte was responsible for macrophage activation. NAC, a reactive oxygen species (ROS) scavenger, was able to ameliorate irinotecan-induced liver injury in vivo. Our data thus provide new insights into the molecular mechanisms of irinotecan-induced liver injury ideas, as well as therapeutic targets, for the development of liver-protective drugs in clinical practice.

## 2. Materials and Methods

### 2.1. Animals and Cell Culture

The experimental protocol and the use of animals were approved by the Biomedical Research Ethics Committee of Hunan Normal University (D2021010). Six-week-old male KM mice were purchased from Hunan Silaike Jingda Laboratory Animal Co., Ltd. The animals were fed in standard laboratory conditions for a week before treated with drugs. KM male mice were randomly divided into 4 groups according to weight, and each group included 4–5 animals. The mice of the liver injury group were intraperitoneally injected with irinotecan hydrochloride (Selleck, S2217, dissolved in 5% glucose solution, Houston, TX, USA), 60 mg/kg, and the control group was injected with 5% glucose solution (Kelun, Chengdu, China) thrice a week on alternate days for 14 days. N-Acetyl-L-cysteine (NAC) (MCE, HY-B0215) was dissolved in 5% glucose solution. The NAC group was injected with IRT (60 mg/kg, i.p.) with NAC (100 mg/kg, i.p.). The macrophage depletion group was injected with IRT (60 mg/kg, i.p.), and mice were intravenously injected (tail vein) with 200 μL clodronate liposomes (Liposoma, C-005, Amsterdam, Netherlands) or control liposome (CL-PBS) on day 7. Mice were sacrificed and liver tissue was fixed with 4% formaldehyde solution or added to RNA later for storage for total RNA extraction.

The normal human liver cells of the L02 type were provided by Professor Xiaoping Yang (Hunan Normal University School of Medicine). The cells were maintained in RPMI1640 (Gibco, Waltham, MA, USA), supplemented with 10% fetal bovine serum (Gibco, USA) and antibiotic-antifungal substance (Basal Media, Shanghai, China) and incubated at 37 °C in humidified air containing at 5% CO_2_.

### 2.2. Antibodies and Chemicals

Antibodies were purchased from the following suppliers: anti-F4/80 (Cell Signaling Technology, Danvers, MA, USA), anti-CD11b (Abcam, Waltham, MA, USA), anti-HMOX-1 (Abcam, Waltham, MA, USA), and anti-caspase 3 (Proteintech, Wuhan, China). Chemicals were purchased from the following suppliers: irinotecan hydrochloride (Selleck, S2217), N-Acetyl-L-cysteine (NAC) (MCE, HY-B0215), clodronate liposomes (C-005), and control liposome (CL-PBS) (Liposoma).

### 2.3. Measurement of Intracellular ROS

L02 cells (3 × 10^5^) and primary mouse hepatocytes (3 × 10^5^) were cultured in a 24-well plate and incubated with 6 μM and 10 μM irinotecan for 24 h, and the levels of intercellular ROS were measured by a microplate reader (BioTek, SYNERGY HTX, Winooski, VT, USA) according to the manufacturer’s introduction. In brief, DCFH-DA was diluted with serum-free culture medium to a final concentration of 10 µmol/L and added to the cells after removing medium and incubated for 20 min at 37 °C. After 3 washings with serum-free medium, fluorescence intensity was measured at 488 nm excitation wavelength and 525 nm emission. For NAC treatment, 600 μM of NAC solution was added, along with the irinotecan. Immunofluorescence images were captured using a confocal microscope (Olympus, FV3000, Tokyo, Japan).

### 2.4. Isolation of Mouse Peritoneal Macrophages

Six to eight week old mice were sacrificed by cervical dissection and immersed in 75% ethanol for 10 s. An amount of 10 mL of PBS was injected intraperitoneally and 5–6 mL of ascites was extracted after gently rubbing the abdomen for 1–2 min. Peritoneal fluid containing resident peritoneal cells were plated into 12-well plates for 1 h at 37 °C. Nonadherent cells were removed by washing (five times) with cold PBS. Macrophages were maintained in RPMI 1640 medium, which includes 10% FBS (Gibco, Billings, MT, USA) and 1% antibiotic-antifungal (Basal Media, Shanghai, China) overnight for further investigation.

### 2.5. Isolation of Primary Mouse Hepatocytes

Primary mouse hepatocytes were isolated by using a multi-step collagenase procedure, as previously described [30]. In brief, the liver was perfused with calcium-free solution and then digested with a collagenase (Sigma-Aldrich, St. Louis MO, USA) perfusion. Dispersed cells were released from the isolated liver, and hepatocytes were collected by 50× *g* centrifugation.

### 2.6. Supernatant Transfer Experiments

In vitro co-culture was set up with primary mouse hepatocytes and peritoneal macrophages. After treating the stably cultured primary mouse hepatocytes with 10 μM irinotecan for 24 h, the supernatant was transferred to the macrophages for another 24 h. Total RNA was then extracted using Trizol for real-time PCR analysis.

### 2.7. Transwell Indirect Co-Culture System

In vitro co-culture with peritoneal macrophages and primary mouse hepatocytes were performed in transwells with a 0.4 μm pole. The peritoneal macrophage was placed in the upper chambers with 10 μM irinotecan and 600 μM NAC was added after the culture was stabilized. The primary hepatocytes were placed at the bottom well and co-cultured for 24 h. The hepatocytes in the lower well plate were used to determine the apoptosis by the TUNEL method. Total RNA was then extracted using Trizol for real-time PCR analysis.

### 2.8. RNA Isolation and Real-Time PCR

Total RNA was isolated from cells using the Trizol reagent (Thermo Fisher Scientific), followed by cDNA synthesis using an RNA reverse transcription kit (Applied Biosystems, Thermo Fisher Scientific, Waltham, MA, USA). Quantification was conducted by qPCR Master Mix (Accurate Biology), and the thermal cycle was as follows: 95 °C for 60 s, 1 cycle; 95 °C for 15 s; and 61 °C for 20 s, 50 cycles. The results were calculated with the 2^−ΔΔCt^ method, with GAPDH serving as the internal reference. Primer sequences are presented in Table 1.

### 2.9. Biochemical Assays

ALT levels were determined using an ALT reagent kit (Nanjing JianCheng Bioengineering Institute, Nanjing, China), and triglyceride level was measured by a triglyceride (TG) level measurement kit (Beijing Boxbio Science & Technology, Beijing, China).

### 2.10. Immunohistochemistry Staining (IHC)

Paraffin slices of liver tissue were dewaxed with xylene and ethanol, then placed in EDTA (1 mmol/L) or sodium citrate solution, boiled in a pressure cooker for 5 min, and then transferred to room temperature for 15–30 min. Then they were incubated with 4% BSA at room temperature for 30 min. Liver tissue slices were incubated with primary antibodies overnight at 4 °C: CD11b (Abcam) and F4/80 (CST). Secondary antibodies were incubated at 37 °C for 30 min. After DAB chromogenic and nucleus counterstaining, the slices were sealed with neutral gum. Images were acquired using a Zeiss Axiolab 5 Digital Lab Microscope (Carl Zeiss AG, Jena, Germany).

### 2.11. TUNEL Measurement of Apoptosis

Experimental manipulations were performed with a TUNEL staining kit (Nanjing Novozymes). Cell samples were fixed with 4% paraformaldehyde for 30 min. Paraffin sections were dewaxed, incubated dropwise with proteinase K for 20 min at 37 °C, and washed 2–3 times with PBS. TUNEL detection solution was added dropwise, incubated at 37 °C for 60 min, washed 3 times with PBS for 5 min each time, DAPI staining solution was added dropwise, incubated at room temperature for 5 min, washed with PBS, sealed with anti-fluorescence quenching solution dropwise, and photographed under fluorescence microscope for observation, and cells were quantified with Image J (National Institute of Health, Bethesda, MD, USA).

### 2.12. Statistical Analysis

Statistical analysis of experimental results was performed with GraphPad Prism 9. All experiments were repeated at least three times, independently, and data were expressed as mean ± SD. One-way ANOVA was used for comparison of means between multiple groups, and the Tukey test was used for two-way comparison within groups. Variance between groups met the assumptions or the appropriate test. Unless otherwise stated, a *p*-value of <0.05 was considered statistically significant.

## 3. Results

### 3.1. Irinotecan Induced Liver Injury in Mice

We first investigated irinotecan-induced liver injury in vivo. Irinotecan was applied to mice by intraperitoneal injection (60 mg/kg every two days) for two weeks, while the control mice were injected with solvent only. We found decreased body weight and body weight-to-liver weight ratio in irinotecan mice when compared with the control group (Figure 1A). Alanine aminotransferase (ALT) was increased in irinotecan-treated mice (Figure 1B). H&E staining of liver sections indicated irinotecan induced massive inflammatory cell infiltration, liver parenchymal damage, and fatty vacuoles in mice (Figure 1C). Correspondingly, triglyceride (TG) levels were significantly increased after irinotecan treatment when compared with control mice (Figure 1D). Consistent with this, qPCR results also indicated that mRNA levels of diglyceride acyltransferase 2 (DGAT2), the enzyme that catalyzes the final step of TG formation, was also significantly unregulated in irinotecan groups compared to the control group (Figure 1E).

### 3.2. Irinotecan Induced the Activation of Macrophage

We further characterized irinotecan-induced liver injury and found that irinotecan administration caused significant elevation of hepatic macrophage marker F4/80, neutrophil marker Ly6G, and monocyte marker CD11b (Figure 2A). Elevated immune cell markers were associated with increased mRNA levels of damage-associated molecular pattern (DAMP) high mobility histone B1 (HMGB-1) and cytokines, including TNF-α, IL-1β, IL-6, and IL10 (Figure 2B). We further confirmed the number of immune cells after irinotecan administration by using immunohistochemistry (IHC), and similar results were observed in liver sections stained with F4/80 and CD11b (Figure 2C). However, Ly6G did not show a significant difference maybe due to the relative late phase of injury (Figure 2C). A macrophage was previously streamlined as pro-inflammatory (M1), and alternative (M2) phenotypes by distinct surface markers were observed [31]. We thus further evaluated the macrophage phenotype after irinotecan treatment by qPCR, and the results indicated that both pro-inflammatory macrophage surface markers iNOS (Figure 2D) and anti-inflammatory macrophage markers Arg-1 and Marco (Figure 2E) were significantly elevated in irinotecan-treated mice. The results of qPCR also showed that the mRNA expression levels of hemoglobin oxygenase-1 (HMOX-1), superoxide dismutase 1 (SOD-1), and glutathione peroxidase 4(GPX-4) was upregulated in mice after irinotecan treatment, suggesting the presence of oxidative stress (Figure 2F). Since HMOX-1 is predominantly expressed in macrophages, we further examined whether HMOX-1 elevation resulted from the irinotecan induced increase of macrophages by using IHC in liver sections (Appendix A). The results indicated that HMOX-1 was predominately expressed in nonparenchymal cells in normal livers. However, after irinotecan treatment, HMOX-1 was increased in hepatocytes (Appendix A).

### 3.3. Macrophage Depletion Prevented Irinotecan-Induced Inflammation and Liver Injury

Macrophage activation is critical in hepatic inflammation and liver injury [20,32], and our data clearly indicated hepatic macrophage activation after irinotecan administration. To examine the role of macrophages in irinotecan-induced liver injury, we treated irinotecan injected mice with clodronate liposomes (CL) to deplete hepatic macrophages on day seven, and we administered control liposomes (CL-PBS) to make sure the effects observed are due to macrophage depletion exclusively. Then, we measured liver injury and inflammation on day 15. There were no significant differences between the IRT and IRT + CL (PBS) groups (Figure 3). However, macrophage depletion prevented irinotecan-induced elevation of ALT (Figure 3A) and partially reversed TG levels (Figure 3B). The qPCR results showed that mRNA levels of F4/80 and CD11b were significantly downregulated after the use of clodronate liposomes (Figure 3C). The IHC results confirmed the significant clearance of macrophages and monocytes after treatment with clodronate liposomes (Figure 3D). H&E staining indicated that macrophage depletion ameliorated irinotecan-induced immune cell infiltration, as well as lipid accumulation (Figure 3E). We further measured mRNA levels, and the results indicated that macrophage depletion significantly suppressed irinotecan-induced elevation of HMGB-1, cytokines, including TNF-α, IL-6, IL-1β, and IL-10, pro-inflammatory macrophage surface markers, iNOS, as well as the anti-inflammatory macrophage makers Arg-1 and Marco (Figure 3F).

Irinotecan promotes TLRs expression in macrophages both in vitro and in vivo.

To investigate mechanisms by which irinotecan activates macrophages, we firstly measured expression of toll-like receptors (TLRs), which are a class of pattern recognition receptors (PRRs) that are normally expressed on proinflammatory cells, such as macrophages and dendritic cells [33] to initiate inflammatory responses by recognizing pathogen-associated molecular patterns (PAMP) and endogenous damage-associated molecular patterns (DAMP) [34]. These mediate the production of various cytokines and chemokines [35]. Our data revealed that TLR1, 4, 5, 6, and 8 were upregulated in mice treated with irinotecan (Figure 4A). To further access whether irinotecan upregulates TLR expression directly, we isolated primary peritoneal macrophages and treated them with irinotecan and examined TLR mRNA levels (Figure 4B). The results indicated that irinotecan was sufficient to upregulate TLR1, 2, 4, 6, and 7 in primary macrophages.

### 3.4. Irinotecan Induced ROS in Hepatocyte Was Responsible for Macrophage Activation

Irinotecan-induced steatohepatitis is accompanied by increased levels of ROS, which are recognized by TLRs and initiate macrophage activation [36,37,38]. We thus tested whether ROS is responsible for macrophage activation in our model. We treated L02 cells and primary mouse hepatocytes with different concentrations of irinotecan for 24 h and detected intracellular ROS level with the reactive oxygen species kit. Irinotecan treatment produced significant elevation of ROS levels in both L02 cells and primary mouse hepatocytes (Appendix A), and this concentration was chosen for subsequent experiments. We used N-acetylcysteine (NAC) as a scavenger of reactive oxygen species to ameliorate oxidative stress [39,40]. We reviewed the literature to determine the effective concentration range in which NAC can eliminate ROS [41,42,43], and we further determined the experimental concentration of NAC by concentration gradient experiments (Appendix A). We found that high concentrations of NAC, together with irinotecan, resulted in a significant decrease in cell numbers (Appendix A). Therefore, we finally determined to use 600 μM of NAC for subsequent experiments. We found that NAC intervention significantly downregulates ROS levels in both L02 cells and primary mouse hepatocytes (Figure 5A,B). We also quantified the fluorescence intensity of ROS after irinotecan and NAC treatment (Figure 5C). These data confirmed that irinotecan induced ROS accumulation in hepatocytes.

To further investigate the role of irinotecan-induced ROS in hepatocytes on macrophage activation, we treated primary macrophages with supernatant from primary hepatocytes with/without treatment of irinotecan and measured macrophage activation by using Arg-1, IL10, IL-6, IL-1β, and TNF-α as markers (Figure 5D). qPCR results showed that irinotecan treatment resulted in significant upregulation of TNF-α and IL-6 while mildly modulating the other markers (Figure 5D). In contrast, treatment with the supernatant of primary hepatocytes treated with irinotecan resulted in more pronounced macrophage activation than irinotecan alone (Figure 5D). Notably, clearing ROS by adding NAC to the supernatant of primary hepatocytes treated with irinotecan completely abolished the activation of macrophages (Figure 5D), while NAC itself had no effects on macrophage activation (Figure 5D). In addition, we treated primary hepatocytes with irinotecan and NAC, but there were no significant changes in the expression of Arg-1, IL-10, IL-6, IL-1β, and TNF-α (Appendix A). We finally tested whether irinotecan-induced ROS in hepatocytes is responsible for TLR upregulation in macrophages (Figure 5E). The mRNA expression levels of TLR-1, TLR-2, TLR-4, TLR-6, and TLR-7 were upregulated after macrophages received the supernatant from irinotecan-treated hepatocytes, and NAC intervention abolished the upregulation of TLR-1, TLR-2, TLR-4, TLR-6, and TLR-7 (Figure 5E).

### 3.5. Irinotecan Induced Macrophage Activation Mediated Lipid Metabolism Disorder and Apoptosis in Hepatocytes

We further investigated the effects of irinotecan-induced macrophage activation on lipid metabolism and apoptosis in primary hepatocytes by using co-culture systems. We first detected indicators related to lipid metabolism in primary hepatocytes after coculturation with macrophages with/without irinotecan by using qRT-PCR. We found that fatty acid synthase (FAS), a key enzyme regulating the pathway of de novo lipogenesis, did not change significantly after irinotecan treatment, whereas the expression level of diglyceride acylase 2 (DGAT2), which catalyzes the final step of TG formation, was significantly increased (Figure 6A). PPAR-α, a major regulator of hepatic lipid metabolism, was downregulated after irinotecan treatment. Meanwhile, irinotecan treatment also downregulated the expression of carnitine palmitoyl transferase IA (CPT-1A), a key enzyme of mitochondrial β-oxidation. In contrast, expression of acyl-coenzyme A oxidase 1 (ACOX1) after irinotecan treatment suggests activation of the extra-mitochondrial fatty acid oxidation system in the peroxisome, which is known to lead to enhanced production of ROS [44] (Figure 6B). NAC prevented irinotecan-induced ACOX-1 elevation, and reversed macrophage activation mediated downregulation of CPT-1A, but showed mild effects on the other factors (Figure 6A,B). We further measured the apoptosis of hepatocytes under co-culture conditions by using the TUNEL assay, and the results indicated that co-culture of THP-1 cells treated with irinotecan with L02 cells significantly induced apoptosis of hepatocytes, which could be prevented by NAC (Figure 6C,E). We repeated these experiments with primary macrophages and the PMH co-culture system and obtained similar results (Figure 6D,F). To further validate the results of in vitro experiments under in vivo conditions, we performed IHC staining of cleaved caspase 3 and TUNEL staining in mouse liver sections. The results showed increased cleaved caspase 3 in irinotecan-treated mice livers (Appendix A). However, we did not observe obvious TUNEL positive cells in the liver maybe due to the early phase of liver injury (Appendix A).

### 3.6. NAC Ameliorates Irinotecan-Induced Liver Injury via Scavenging ROS under In Vivo Conditions

We finally investigated whether NAC could prevent irinotecan-induced liver injury in vivo. Mice were injected with IRT (60 mg/kg, i.p.) in the presence or absence of NAC (100 mg/kg, i.p.) for two weeks. We found that the ALT level, TG content, and expression of DGAT-2 in the NAC group were significantly lower than those in the irinotecan-injured group, but TG content and the expression of DGAT-2 were still higher than in the control group (Figure 7A). Irinotecan treatment caused a marked increase of heme oxygenase-1 (HMOX-1) expression, and NAC treatment reversed it (Figure 7B). More interestingly, NAC treatment decreased irinotecan-induced HMOX1 elevation in hepatocytes, but mildly impacted non-parenchymal cell staining (Appendix A). Consistent with the results of in vitro cellular assays, irinotecan did not change FAS expression, but significantly inhibited PPAR-α expression, and the role of NAC in resisting this effect was not significant (Figure 7C). mRNA levels of F4/80 and CD11b were downregulated after NAC treatment, and the mRNA levels of Arg-1, MARCO, iNOS, TNF-α, IL-6, IL-1β, IL-10, and HMGB1 were downregulated to different degrees (Figure 7D). H&E staining results showed that the infiltration of inflammatory cells was improved after NAC treatment (Figure 7E). Immunohistochemical staining showed that F4/80 and CD11b positive cells in the livers of NAC-treated mice were lower than those in the irinotecan groups (Figure 7F). We also found that NAC treatment abolished irinotecan-induced upregulation of TLR expression (Figure 7G), and it also increased cleaved caspase 3 in the liver (Appendix A).

## 4. Discussion

In this study, we demonstrated that irinotecan administration caused significant elevation of ALT, inflammation, and fat accumulation in the liver. Depletion of macrophages prevented irinotecan-induced liver injury and inflammatory response in mice. In vitro data indicated that irinotecan induced intracellular ROS production in primary hepatocytes and upregulated TLR family expression in macrophages. Supernatant from irinotecan-treated hepatocytes triggered macrophage activation and upregulation of TLRs in macrophages, and NAC abolished these effects. We further revealed that irinotecan activated macrophage-induced impairment of lipid metabolism and promoted apoptosis in hepatocytes, and NAC was able to prevent macrophage-induced cell death and partially reverse impaired lipid metabolism. Finally, we demonstrated that combining NAC with irinotecan prevented irinotecan-induced macrophage activation, TLR upregulation, liver injury, and partially reversed the accumulation of triglycerides in vivo. Our results indicated that macrophages play critical roles in irinotecan-induced liver injury, and targeting ROS provides new options for the development of hepatoprotective drugs in clinical practice.

Macrophages are a class of phagocytes with self-renewal capacity and are sentinels of liver homeostasis [23]. Macrophage activation is critical for liver disease, including alcohol-related liver disease (ALRD), fibrosis, drug-induced liver injury (DILI), as well as liver cancer [45,46], by initiating inflammation, tissue repair, and immune regulatory effects [47]. Modulating macrophage activation offers novel therapeutic targets for both acute and chronic liver disease [48,49]. In line with those observations, we found that macrophage activation is associated with irinotecan-induced liver injury, and the depletion of macrophages completely prevents liver injury caused by irinotecan. Our data thus add new mechanisms underlying irinotecan-induced liver injury. Of note, macrophages are extremely plastic and can adjust their phenotype according to signals from the liver microenvironment, including nutrients, metabolites, and oxygen, which result in significant diversity of the macrophage population [50]. The diversity of macrophages was previously streamlined as a simplified concept of M1 and M2 phenotypes by distinct surface markers [51,52]. In our experiments, we observed that both M1 and M2 types of macrophage markers were elevated after irinotecan treatment. This indicated the complexity and plasticity of macrophages after irinotecan treatment.

Macrophages mediate the immune response by recognizing substances, such as danger signaling molecules, fatty acids, cellular debris, and ROS [53,54,55]. Macrophage activation is mainly achieved through recognition of surface pattern recognition receptors, including TLRs [56]. Our data indicated that irinotecan upregulates TLRs in macrophages through direct and indirect mechanisms. On one hand, irinotecan treatment directly stimulates TLR expression, and it triggers ROS production in hepatocytes, which also contributes to TLR upregulation. Notably, NAC treatment in a co-culture system and in and in vivo experiment abolished irinotecan-induced TLR upregulation, suggesting ROS plays more profound roles in mediating TLR expression. However, the specific mechanism by which irinotecan directly upregulates TLRs expression is still unclear, but our data provided new evidence underlying irinotecan-induced gastrointestinal toxicity and hepatic injury [57,58].

By using a hepatocyte/macrophage co-culture system, we revealed that activated macrophages caused disruption of hepatocyte lipid metabolism and apoptosis. Previous studies have shown that reactive oxygen species-mediated oxidative stress leads to disturbed hepatic lipid metabolism, which is an important mechanism leading to liver disease [59]. Meanwhile, both TNF-α and IL-1β directly inhibit the activation of PPARα to upregulate FAS expression [60,61]. TNF-α also inhibits β-oxidation by inhibiting peroxisomal fatty acyl-CoA oxidase, which promotes hepatic steatosis [62]. In our results, ROS clearance prevented macrophage-induced hepatocyte apoptosis, but only partially reversed impairment of lipid metabolism in hepatocyte. The crosstalk between macrophages and hepatocytes still needs further investigation, but we speculate that irinotecan causes hepatic lipid metabolism disorders by inducing pro-inflammatory cytokines and ROS produced by macrophages.

N-acetylcysteine (NAC) is a commonly used reactive oxygen scavenger for the clinical treatment of hepatotoxicity caused by acetaminophen (APAP) overdose [63]. Our results showed that NAC could scavenge oxidative stress, lower ALT levels, and reduce the release of inflammatory factors in mice. Although NAC improved the infiltration of inflammatory cells, the improvement of lipid metabolism did not seem to be obvious. At present, NAC is rarely used for the treatment of liver injury induced by chemotherapy drugs, and the combination itself will impose a certain metabolic burden on the patient’s liver [64,65]. Whether NAC can be an adjuvant drug for the treatment of liver injury induced by chemotherapy drugs still needs to be further investigated in clinical trials.

## Figures and Tables

**Figure 1 cells-11-03791-f001:**
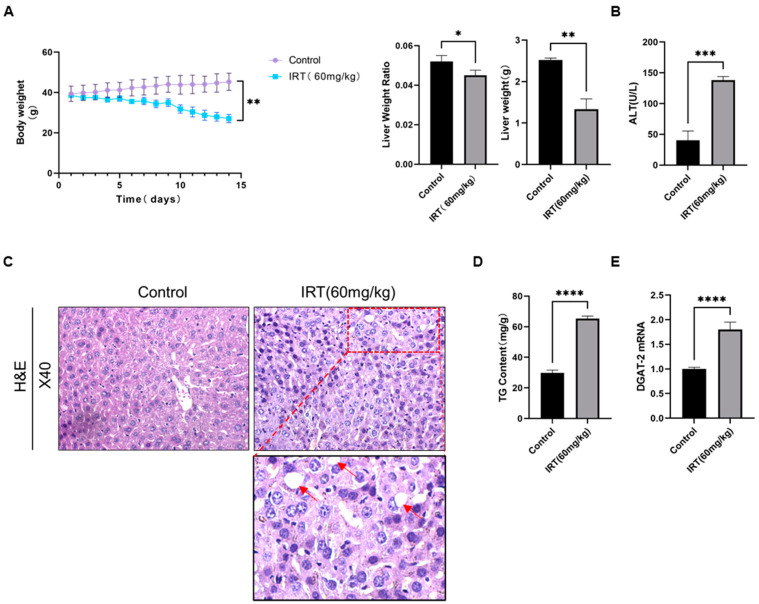
Irinotecan-induced liver injury in mice. Mice were intraperitoneally injected with irinotecan hydrochloride (60 mg/kg) or 5% glucose solution thrice a week on alternate days for 14 days. (**A**) Changes in body weight, liver weight, and liver-to-body weight ratio of mice after irinotecan treatment. (**B**) Serum levels of alanine aminotransferase (ALT) in mice. (**C**) Representative images of H&E staining from liver tissues of irinotecan-treated mice and control mice. The appearance of vacuoles is indicated by arrows. (**D**) Analysis of triglyceride (TG) content in liver tissues of control and irinotecan-treated mice. (**E**) mRNA level of diglyceride acyltransferase 2 (DGAT2). Data are reported as mean ± SD, and each graph is representative of at least three independent experiments. Statistical significance was tested with an unpaired *t*-test. * *p* < 0.05; ** *p* < 0.01; *** *p* < 0.001; **** *p* < 0.0001.

**Figure 2 cells-11-03791-f002:**
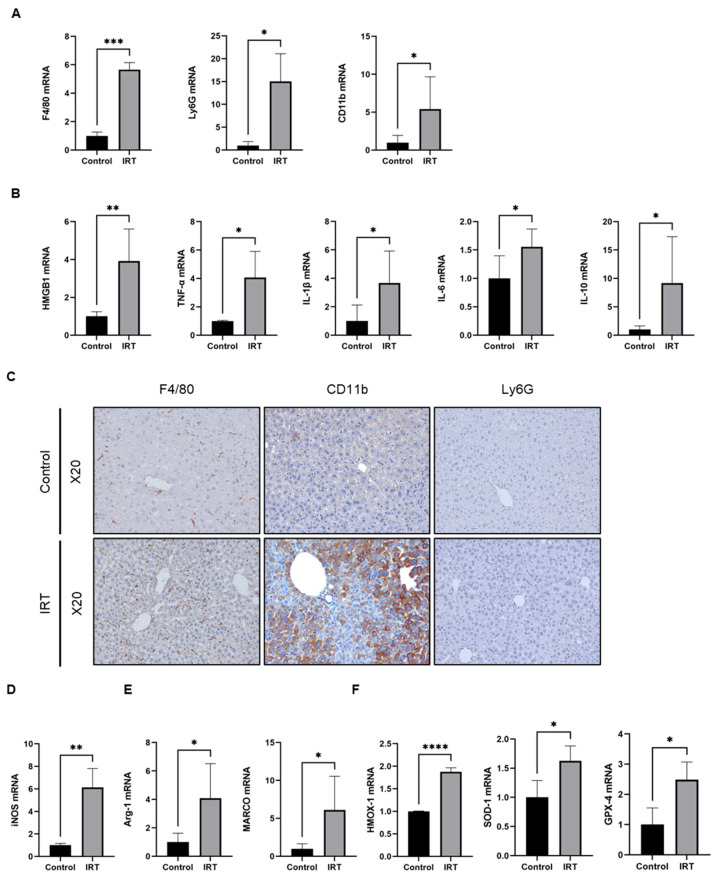
Irinotecan induced activation of macrophages. Mice were intraperitoneally injected with irinotecan hydrochloride (60 mg/kg) or 5% glucose solution thrice a week on alternate days for 14 days. (**A**) Hepatic mRNA Expression levels of F4/80, Ly6G, and CD11b after two weeks of irinotecan treatment. (**B**) Hepatic mRNA Expression levels of HMGB1, TNF-α, IL-1β, IL-6, and IL-10. (**C**) Representative images of immunohistochemical staining of F4/80, CD11b, and Ly6G in mouse liver tissue. (**D**,**E**) Hepatic mRNA Expression of pro-inflammatory macrophage surface marker iNOS (**D**) and anti-inflammatory macrophage surface markers Arg-1 and MARCO (E) after irinotecan treatment in mice livers. (**F**) Changes in mRNA expression levels of HMOX-1, SOD-1 and GPX-4 in mice after irinotecan treatment. Data are reported as mean ± SD, and each graph is representative of at least three independent experiments. Statistical significance was tested with an unpaired *t*-test. * *p* < 0.05; ** *p* < 0.01; *** *p* < 0.001; **** *p* < 0.0001.

**Figure 3 cells-11-03791-f003:**
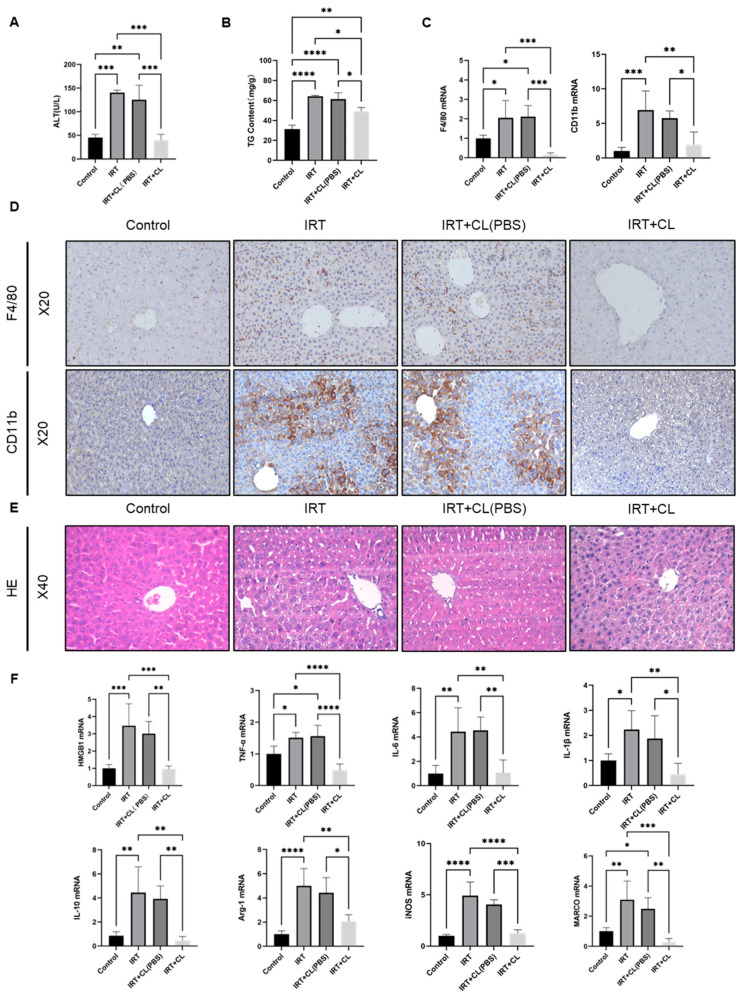
Macrophages play a critical role in irinotecan-induced liver injury. Mice were intraperitoneally injected with irinotecan hydrochloride (60 mg/kg) and treated with 200 uL of clodronate liposomes (CL) and control liposomes (CL-PBS) through tail vein injection on day seven, and liver injury was measured on day 15. Serum levels of ALT (**A**) and TG content (**B**) in mice after irinotecan treatment and CL intervention. (**C**) mRNA expression levels of F4/80 and CD11b after CL and CL (PBS) intervention. (**D**) Representative images of immunohistochemical staining for F4/80 and CD11b after irinotecan treatment and CL or CL (PBS) intervention. (**E**) H&E staining of liver sections from mice after irinotecan treatment and CL or CL (PBS) intervention. (**F**) mRNA expression of HMGB1, TNF-α, IL-6, IL-1β, IL-10, Arg-1, iNOS, and MARCO in mice after irinotecan treatment and CL or CL (PBS) intervention. Data are reported as mean ± SD, and each graph is representative of at least three independent experiments. Comparison of values was performed by one-way ANOVA for unpaired data. * *p* < 0.05; ** *p* < 0.01; *** *p* < 0.001; **** *p* < 0.0001.

**Figure 4 cells-11-03791-f004:**
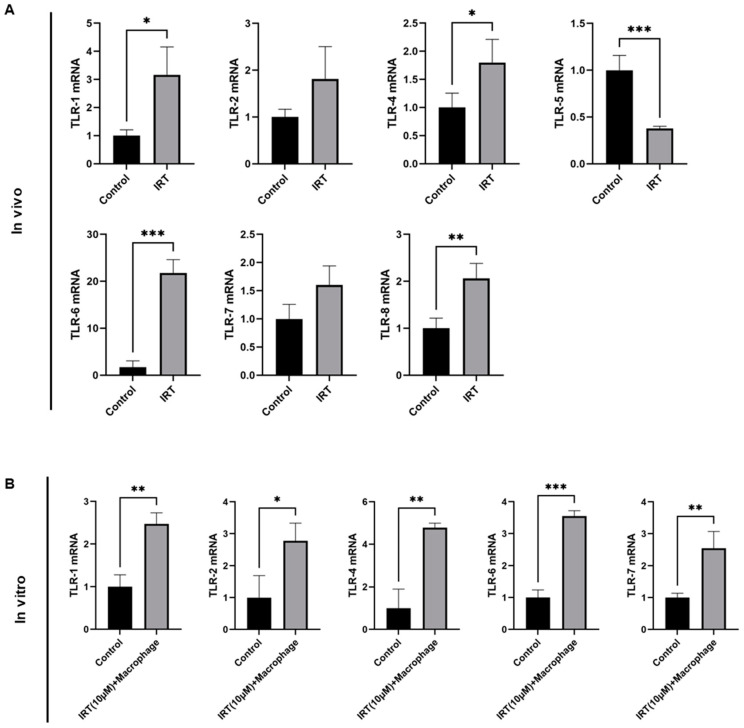
Irinotecan promotes upregulation of TLR family expression. In in vitro experiments, total RNA was extracted for subsequent qPCR experiments after treating macrophages with 10 μM of irinotecan for 24 h. (**A**) Changes in TLR family mRNA expression levels in mice livers after irinotecan treatment. (**B**) Primary peritoneal macrophages were treated with 10 μM irinotecan for 24 h, and TLR family mRNA expressions were measured by qPCR. Data are reported as mean ± SD, and each graph is representative of at least three independent experiments. Comparison of values was performed by one-way ANOVA for unpaired data. * *p* < 0.05; ** *p* < 0.01; *** *p* < 0.001.

**Figure 5 cells-11-03791-f005:**
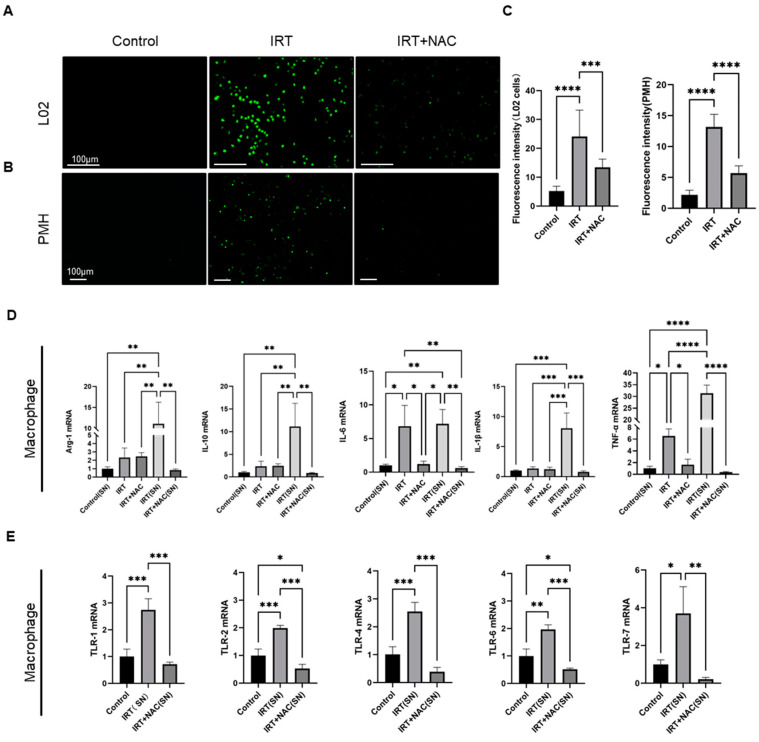
Irinotecan activates macrophages and TLRs family expression through ROS production in hepatocytes. L02 or primary mouse hepatocytes (PMH) were treated with irinotecan (6 μM and 10 μM, respectively) in the absence or presence of NAC (600 μM) for 24 h, and the changes of intracellular ROS content were detected, and the supernatant was transferred to macrophages and continued to be cultured for 24 h, and the total RNA of macrophages was extracted for subsequent qPCR experiments. Determination of ROS activity in L02 (**A**) and PMH cells (**B**) after irinotecan and NAC treatment. (**C**) Quantitative analysis of fluorescence intensity of reactive oxygen species in L02 cells and PMH after treatment with irinotecan and NAC. (**D**) Changes in mRNA levels of Arg-1, IL-10, IL-6, IL-1β, and TNF-α in the control (SN) group, the irinotecan direct treatment group, the irinotecan (SN) treatment group, the irinotecan and NAC direct treatment group, and the irinotecan and NAC (SN) treatment group. (**E**) Changes in mRNA levels of the TLR family in macrophages treated with supernatants from primary hepatocytes treated with irinotecan (10 μM) in absence or presence of NAC (600 μM). Data are reported as mean ± SD, and each graph is representative of at least three independent experiments. Statistical significance was tested with an unpaired *t*-test and one-way ANOVA. * *p* < 0.05; ** *p* < 0.01; *** *p* < 0.001; **** *p* < 0.0001.

**Figure 6 cells-11-03791-f006:**
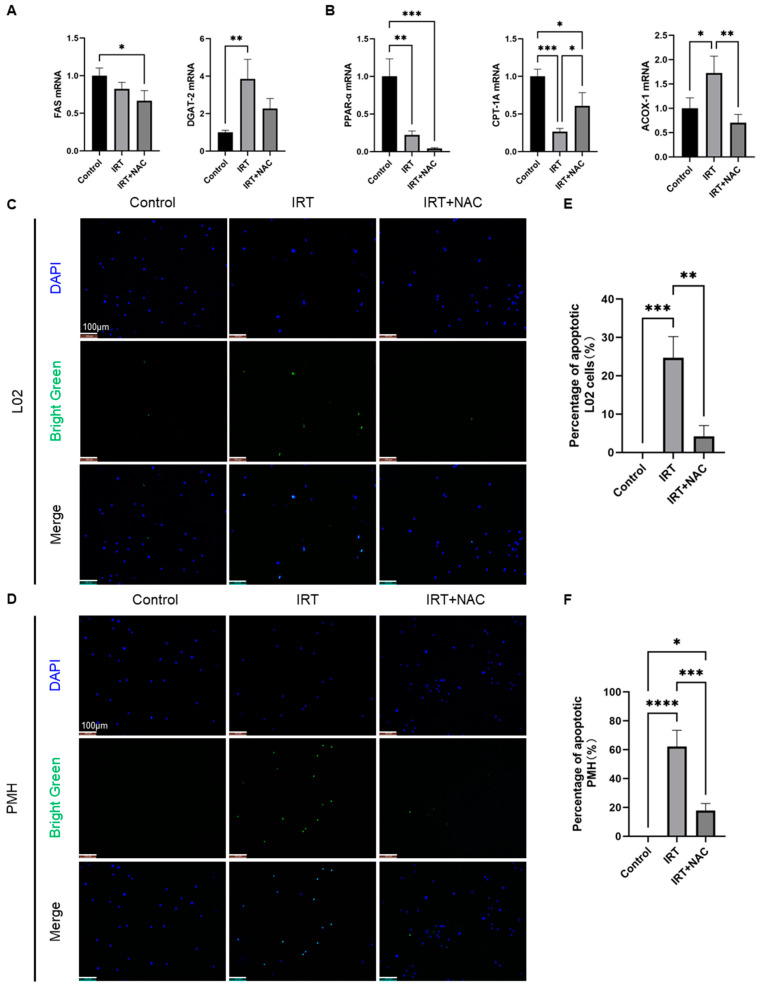
Irinotecan induced macrophage activation trigger lipid metabolism disorder and apoptosis in hepatocytes. (**A**,**B**) The macrophages were treated with irinotecan (10 μM) in the absence or presence of NAC (600 μM), and they were indirectly co-cultured with hepatocytes, and the lower hepatocytes were used for TUNEL or qPCR. (**A**) Expression of mRNA levels of FAS and DGAT-2 in hepatocyte after NAC intervention. (**B**) Expression of mRNA levels of PPAR-α, CPT-1A, and ACOX-1 in hepatocytes after NAC intervention. (**C**–**F**) L02 cells (**C**) or PMH (**D**) were co-cultured with THP-1 (**C**) or primary macrophages (**D**) treated with irinotecan (10 μM) in the absence or presence of NAC (600 μM). Cell death was evaluated by TUNEL assay. Quantitative analysis of changes in the percentage of apoptotic cells after irinotecan treatment and NAC intervention, respectively (**E**,**F**). Data are reported as mean ± SD, and each graph is representative of at least three independent experiments. Statistical significance was tested with an unpaired *t*-test and one-way ANOVA. * *p* < 0.05; ** *p* < 0.01; *** *p* < 0.001; **** *p* < 0.0001.

**Figure 7 cells-11-03791-f007:**
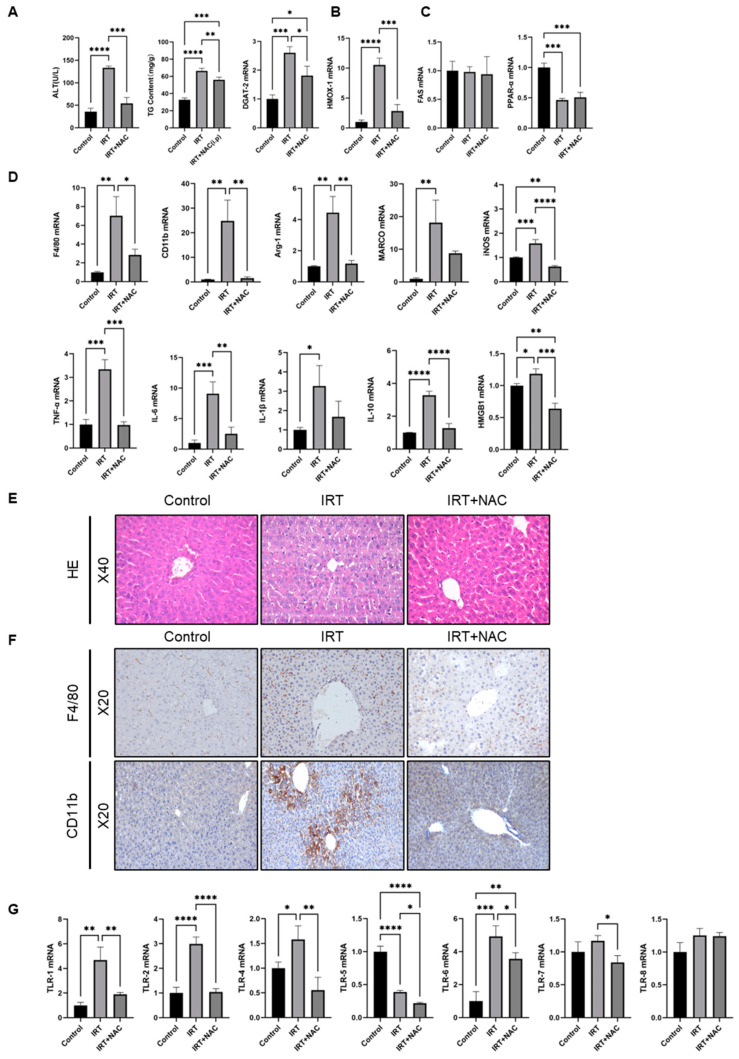
NAC ameliorates irinotecan-induced liver injury under in vivo conditions. Mice in the IRT group were injected intraperitoneally with irinotecan hydrochloride (60 mg/kg), and mice in the NAC group were simultaneously injected intraperitoneally with NAC (100 mg/kg) on the following day, alternately for 2 weeks, and liver injury was measured on day 15. (**A**) Changes in ALT levels, TG content, and mRNA expression of DGAT-2 in mice after NAC intervention. (**B**) Changes in mRNA expression of HMOX-1 in mice after NAC intervention. (**C**) Changes in mRNA expression of FAS and PPAR-α in mice after NAC intervention. (**D**) Changes in mRNA expression of F4/80, CD11b, Arg-1, MARCO, iNOS, TNF-α, IL-6, IL-1β, IL-10, and HMGB1 in mice after irinotecan treatment and NAC intervention. (**E**) H&E staining of mouse liver after NAC intervention. (**F**) Immunohistochemical staining of F4/80 and CD11b in mouse livers after NAC intervention. (**G**) Changes in mRNA expression of TLRs in mice after NAC intervention. Data are reported as mean ± SD, and each graph is representative of at least three independent experiments. Comparison of values was performed by One-way ANOVA for unpaired data. * *p* < 0.05; ** *p* < 0.01; *** *p* < 0.001; **** *p* < 0.0001.

**Table 1 cells-11-03791-t001:** Primer sequences used for RT-PCR.

Primer	Forward (5′-3′)	Reverse (5′-3′)
TLR-1	TGACCTGCCCTGGTATGTGAG	GGCAGAATCATGCCCACTGTA
TLR-2	GAGCATCCGAATTGCATCACC	CCCAGAAGCATCACATGACAGAG
TLR-4	CATGGATCAGAAACTCAGCAAAGTC	CATGCCATGCCTTGTCTTCA
TLR-5	GCTTGGAACATATGCCAGACACA	AAAGGCTATCCTGCCGTCTGAA
TLR-6	AATGGTACCGTCAGTGCTGGAAATA	TGGCTCATGTTGCAGAGGCTA
TLR-7	CTTTGCAACTGTGATGCTGTGTG	ACCTTTGTGTGCTCCTGGACCTA
TLR-8	ACGGCTTGCCATCTTGGTC	AGTGGCAAATGTTCTTAGGGATTGA
F4/80	CTTTGGCTATGGGCTTCCAGTC	GCAAGGAGGACAGAGTTTATCGTG
CD11b	AAACCACAGTCCCGCAGAGA	CGTGTTCACCAGCTGGCTTA
Ly6G	TGCGTTGCTCTGGAGATAGA	CAGAGTAGTGGGGCAGATGG
iNOS	AATCTTGGAGCGAGTTGTGG	CAGGAAGTAGGTGAGGGCTTG
TNF-α	AGGCTCTGGAGAACAGCACAT	TGGCTTCTCTTCCTGCACCAAA
Arg-1	CTCCAAGCCAAAGTCCTTAGAG	AGGAGCTGTCATTAGGGACATC
Marco	GCACAGAAGACAGAGCCGAT	AGTGATCCATTGCCACAGCA
IL-6	TTCCATCCAGTTGCCTTCTT	CAGAATTGCCATTGCACAAC
IL-β	GGATGAGGACATGAGCACCT	AGCTCATATGGGTCCGACAG
IL-10	GGTTGCCAAGCCTTATCGGA	ACCTGCTCCACTGCCTTGCT
HMGB-1	CGCGGAGGAAAATCAACTAA	GCAGACATGGTCTTCCACCT
HMOX-1	AGGTACACATCCAAGCCGAGA	CATCACCAGCTTAAAGCCTTCT
FAS	CTGCGGAAACTTCAGGAAATG	GGTTCGGAATGCTATCCAGG
DGAT-2	CTGGCTGGCATTTGACT	TCTATGGTGTCTCGGTTGA
PPAR-α	TATTCGGCTGAAGCTGGTGTAC	CTGGCATTTGTTCCGGTTCT
CPT-1A	GCTGCACTCCTGGAAGAAGA	GGAGGGGTCCACTTTGGTAT
ACOX-1	GAGCTGCTCACAGTGACTCG	ACTGCAGGGGCTTCAAGTG
GPX-4	CCACGCAGCCGTTCTTAT	GAGGCAGGAGCCAGGAAGT
SOD-1	TTTTTGCGCGGTCCTTTCCTG	GGTTCACCGCTTGCCTTCTGCT
GAPDH	CGTCCCGTAGACAAAATGGT	TTGAGGTCAATGAAGGGGTC

## Data Availability

All data are within the manuscript and Appendix A. Any additional information or data are available upon request.

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
