# Peer review of "Hepatic ROS Mediated Macrophage Activation Is Responsible for Irinotecan Induced Liver Injury"

_cells, 2022, doi:10.3390/cells11233791_

Round 1
Reviewer 1 Report
This current manuscript could be considered after major revision. Some questions must be addressed.
(1) Please improve the English-writing, there are many grammar mistakes in the current version.
(2) Biochemical assays should be included in the 2. Material and Methods Part.
(3) Author should provide the CL control group.
(4) We know that L02 cells used may be derived from HeLa cells. Therefore, please provided the identification.
(5) NAC at 6 μM should not work. This dose is too low and it is not possible. In fact, the dose of NAC is at 1-0 mM. Please check.
(6) For the cell experiment, author should perform AV/PI staining and run it using flow cytometer. The TUNEL staining should be also performed in the animal study.
Reviewer 2 Report
1.Why KM mice was chosed in vivo experiment? The representation of 4-5 mice in each group is limited.
2.What is the basis for choosing the dose of irinotecan hydrochloride?
3.What is the difference between P<0.001, P<0.001 and P<0.01? ****P<0.0001 should be added in figure 2,5,6,7.
4."In line with those observations, we found that macrophage activation is associated with irinotecan induced liver injury and depletion of macrophage completely prevent liver injury cause by irinotecan." Does depletion of macrophage has any other effect on liver ?
Reviewer 3 Report
The study is very interesting and well-conducted.
I would have though some comments/remarks:
Major comment:
About the in vivo experiments: HMOX1 gene expression should not be the only readout to affirm ROS overproduction after treatment with irinotecan, especially that HMOX1 is highly expressed in liver macrophages in general. It might be interesting to look at some other gene expression of enzymes involved in oxidative stress response and also look at the GSH content.
Also, the induction of HMOX1 expression in the liver might be due to the only presence of macrophages in treated mice. Did you immunostain for HMOX1 to see if HMOX1 was expressed by hepatocytes?
Minor comments:
- Fig.1-C: vacuoles presence is mentioned. It would be great if you could include better images/higher magnification images to be able to see those features.
- Fig.1-D: intrahepatic TG or circulating?
- Fig.1-E: by looking at the expression of DGAT2 after treatment, you assume that the accumulation of TG would be due to activation of de novo lipogenesis. Some other mechanisms could also be responsible for TG accumulation, such as inhibition of mitochondrial FAO for example. Did you look at some gene expression of genes involved in FAO? It might be interesting to add them.
- Fig. 5-A,B; Fig.6-C,D: please, enlarge or add bar size to photographs.
- Also, can you show and merge DAPI/Hoechst staining for both L02 and PMH cells?
- In the body text, replace 'new lipogenesis' by 'de novo lipogenesis'
- Fig.6-C: it seems that some of the DAPI pictures are in fact merged with Tunel. Please, show each channel + merge
Round 2
Reviewer 1 Report
(1) Please STR analysis of L02 cells. I think it is required.
(2) In the part 2.7, i.e., line 214, authors wrote that the dose of NAC is 6 μM. Its dose should be at the mM levels. Indeed, in the published paper, including ourselves, the dose is at the mM levels.
(3) The IHC need the negative control. It looks like author did not perform it.
(4) In 2.11, the detail information of TUNEL is required.
(5)The primary cell isolation should provide data of cell identification.
(6) In the figure6, in L02 cells, the apoptotic rates is about 30%, but i noted there was total 10 cells in the images from IRT group. It is less than that in the control group. But the apoptotic rates only had 30%. It means apoptosis is not the main pathway? Please explain..
